



# Representation of the Equatorial Stratopause Semiannual Oscillation in Global Atmospheric Reanalyses

Yoshio Kawatani[1], Toshihiko Hirooka[2], Kevin Hamilton[3,4], Anne K. Smith[5], Masatomo Fujiwara[6]

[1]Japan Agency for Marine-Earth Science and Technology, Yokohama, 236-0001, Japan
[2]Faculty of Science, Kyushu University, Fukuoka, 819-0395, Japan
[3]International Pacific Research Center, University of Hawaii, Honolulu, 96822, USA
[4]Department of Atmospheric Sciences, University of Hawaii, Honolulu, 96822, USA
[5]National Center for Atmospheric Research, Boulder, 80307, USA
[6]Faculty of Environmental Earth Science, Hokkaido University, Sapporo, 060-0810, Japan

*Correspondence to*: Y. Kawatani (yoskawatani@jamstec.go.jp)

**Abstract.** This paper reports on a project to compare the representation of the semiannual oscillation (SAO) in the equatorial stratosphere and lower mesosphere among six major global atmospheric reanalysis datasets and with recent satellite SABER and MLS observations. All reanalyses have a good representation of the quasi-biennial oscillation (QBO) in the equatorial lower and middle stratosphere and each displays a clear SAO centered near the stratopause. However, the differences among reanalyses are much more substantial in the SAO region than in the QBO dominated region. The degree of disagreement among the reanalyses is characterized by the standard deviation (SD) of the monthly-mean zonal wind and temperature; this depends on latitude, longitude, height, and time. The zonal wind SD displays a prominent equatorial maximum that increases with height, while the temperature SD is minimum near the equator and largest in the polar regions. Along the equator the zonal wind SD is smallest around the longitude of Singapore where consistently high-quality near-equatorial radiosonde observations are available. Interestingly the near-Singapore minimum in SD is evident to at least ~3 hPa, i.e. considerably higher than the usual ~10 hPa ceiling for *in situ* radiosonde observations. Our measurement of the agreement among the reanalyses shows systematic improvement over the period considered (1980–2016), up to near the stratopause. Characteristics of the SAO at 1 hPa, such as its detailed time variation and the displacement off the equator of the zonal wind SAO amplitude maximum, differ significantly among the reanalyses. Disagreement among the reanalyses becomes still greater above 1 hPa. One of the reanalyses in our study also has a version produced without assimilating satellite observations and a comparison of the SAO in these two versions demonstrates the very great importance of satellite derived temperatures in the realistic analysis of the tropical upper stratospheric circulation.

## 1 Introduction

The semiannual oscillation (SAO) is an alternation of the equatorial zonal wind between easterlies and westerlies with a period of six months and is observed from the upper stratosphere to above the mesosphere. The SAO was first detected in rocketsonde observations of the zonal wind near the equator (Reed, 1965, 1966; Hopkins, 1975) and these observations





indicate that the SAO amplitude has two peaks: one near the stratopause (~1 hPa) and the other near the mesopause (~0.01 hPa; Hirota, 1978; 1980). Interest in the present paper will be focused on the stratopause SAO. Below the SAO region the mean wind variations are dominated by the stratospheric quasi-biennial oscillation (QBO). The QBO and SAO zonal wind variations have some similarities, notably a consistent downward progression of the wind reversals and the formation of

strong vertical shear zones. The peak SAO harmonic amplitude determined by Hirota (1978) from rocket observations is over 30 m-s$^{-1}$, which is somewhat larger than the corresponding peak QBO amplitude. Unlike the QBO, which has exhibited periods among individual cycles ranging from ~18–34 months, the SAO is clearly locked to the seasonal cycle. Results from simple models (Dunkerton, 1979; Holton and Wehrbein, 1980), diagnostic studies (Hamilton, 1986; Hitchman and Leovy, 1986; Ray et al., 1998) and comprehensive general circulation models (Hamilton and Mahlman, 1988; Jackson and Gray,

1994) have shown that the SAO is  driven by a combination of cross-equatorial meridional advection of mean momentum, the transports of zonal momentum by vertically propagating equatorial and gravity waves and the wave forcing from extratropical quasistationary planetary waves.

Observations of the winds and temperatures in the upper stratosphere and lower mesosphere are limited compared with the lower and middle stratosphere. There are ~220 radiosonde stations within 10° S–10° N included in the Integrated Global

Radiosonde Archive (IGRA; Durre et al., 2006; NCDC, 2016), but only a fraction of these stations reported many observations in the stratosphere. The number of radiosonde observations decreases significantly with height (Figs. 5 and 15 of Kawatani et al. 2016) and in the IGRA data base there are no observations above the usual 10 hPa ceiling for weather balloons.  Rocketsondes can provide *in situ* measurements of the wind and temperature in the upper stratosphere and lower mesosphere but these observations are available only for a few locations and for limited periods (Hirota, 1980; Garcia et al.,

1997; Baldwin and Gray, 2005).

A unique opportunity to observe global winds in the middle atmosphere was provided by the High Resolution Doppler Imager (HRDI) on the Upper Atmosphere Research Satellite (UARS) during 1992-1996, but the HRDI data are not accurate in the 40-60 km range (Garcia et al., 1997; Ray et al., 1998).  The stratopause region is also hard to observe with ground-based radars as it is above the usual ceiling for atmospheric profilers and below the region that can be observed with meteor

winds and medium-frequency radar techniques.

The temperature in the stratopause region has been observed by satellite based radiometers since 1979 and for about the last two decades there have been specialized limb-sounding instruments deployed that provide higher quality temperature retrievals in the middle atmosphere. Recently, Smith et al. (2017) derived the zonal mean zonal winds by the balance wind relationship using the geopotentials derived from the temperature retrievals from the Thermosphere, Ionosphere, Mesosphere

Energetic and Dynamics (TIMED) Sounding of the Atmosphere Using Broadband Emission Radiometry (SABER) instrument for 15 years and the Aura Microwave Limb Sounder (MLS) for 12.5 years. These datasets have the advantages of long data records with no gaps and continuous vertical coverage through the middle atmosphere. As the balanced gradient





wind equation is not valid near the equator, Smith et al. (2017) estimated equatorial wind by the cubic spline interpolation of the balance winds at and poleward of latitude ± 8° for SABER and ±6° for MLS. They assessed the reliability of their estimations in the lower stratosphere and upper mesosphere zonal wind by comparing with *in situ* radiosonde observations in the lower stratosphere and radar measured meteor winds in the upper mesosphere at Ascension Island (8° S).

Global atmospheric analyses that assimilate all available satellite remote sensing and *in situ* observations are another potential source of information regarding the SAO. Even in assimilating the same observations, differences in the forecast model and data assimilation technique of various observational datasets will lead to differences in their representations of atmospheric fields, including those of the mean state, variability, and long-term trend. Kawatani et al. (2016) compared the representation of the monthly-mean zonal wind in the equatorial stratosphere up to 10 hPa among major global atmospheric
reanalysis datasets. It was found that differences among reanalyses in the zonal wind depend significantly on the number of *in situ* radiosonde observations, the QBO phase, and the representation of extratropical quasi-stationary planetary waves propagating toward the equator.

Kawatani et al. (2016) noted global meteorological analysis processes are particularly challenging in the tropical middle atmosphere, even in the lower stratosphere. This can be attributed to the following: the relative paucity of *in situ* data
(especially in the eastern and central Pacific area with few stations, see their Fig. 5), the weaker constraint connecting the winds and temperatures because of the small Coriolis parameter near the equator, and the relatively coarse vertical resolution of satellite remote-sensing temperature retrievals compared to the thin regions of large vertical shear of the zonal wind characteristic of both the QBO and SAO. As noted earlier, the lack of observations is even more pronounced in the stratosphere above 10 hPa and so in the SAO region we can perhaps anticipate more dependence on the dynamical models
used in the assimilation procedure.

The present paper reports on one of the studies contributing to the SPARC Reanalysis Intercomparison Project (S-RIP; Fujiwara et al., 2017), a project which focuses on evaluating reanalysis output for the middle atmosphere. We compare the representation of the near stratopause SAO by several contemporary reanalysis products and further evaluate the reanalyses by comparing with the Smith et al. (2017) winds derived more directly from satellite temperature observations. We have
restricted our attention to the period starting in 1979, when NOAA operational satellite radiance observations became available and were incorporated as an important data source in all reanalyses. We do not focus on the driving mechanism of the SAO in reanalyses. Tomikawa et al. is now investigating the momentum budgets and compare the mechanism among reanalyses, which is also one of the SRIP papers (personal communication).

Detailed information such as assimilated satellite datasets used in each reanalysis were provided by the S-RIP project,
notably summarized in Fujiwara et al. (2017) and Wright et al. (https://jonathonwright.github.io/pdf/S-RIPChapter2E.pdf). However, as discussed in Kawatani et al. (2016), it is not feasible to determine exactly what observational data were actually



assimilated at each data assimilation analysis step; these complications make it difficult to conclusively attribute all the differences seen among the reanalyses products. In the SAO altitudes, observational data available to be assimilated are particularly limited. In addition, as described in section 2, the treatments of sponge layers near the model top are also different among reanalyses; these factors may be expected to result in significantly larger differences among reanalyses

compared to those from QBO altitudes described by Kawatani et al. (2016). This paper will discuss the results of our detailed intercomparison and will help identify the uncertainties in the reanalyses and how uncertainties change with time as the satellite data sources evolve.

One of the reanalysis datasets considered in our project, the JRA-55 reanalyses produced by the Japan Meteorological Agency, also has a version produced without assimilating satellite observations (JRA-55C). Kobayashi et al. (2014)

compared the JRA-55 and JRA-55C reanalyses and found that the tropical zonal wind difference between the JRA-55 and JRA-55C is larger in the upper stratosphere than in the lower stratosphere. The present paper will include a more detailed comparison between JRA-55 and JRA-55C, allowing a direct assessment of the importance of satellite derived temperatures in the realistic analysis of the tropical upper stratospheric circulation.

This paper is outlined as follows: Section 2 briefly describes the reanalysis products evaluated and the satellite observation

data employed; Section 3 investigates differences in the overall patterns of tropical zonal wind and temperatures among reanalyses; Section 4 discusses the similarities and differences of the SAO among the reanalyses; conclusions are summarized in Section 5.

## 2 Reanalysis and satellite observation data

We analyzed the monthly mean zonal wind and temperature in six sets of global reanalyses data, which are available through

at least 2010 and extend at least up to 1 hPa altitude. Relatively old reanalyses used in Kawatani et al. (2016), i.e. NCEP1, NCEP2, ERA-40, and JRA-25, were not analyzed here; instead, we analyze ERA-I (Dee et al., 2011), ERA5 (Hersbach et al., 2019), JRA-55 (Kobayashi et al., 2015), MERRA (Rienecker et al., 2011), MERRA-2 (Gelaro et al., 2017) and NCEP-CFSR (Saha et al., 2010). To assess the contribution of satellite observations, we also analyze JRA-55C, which assimilated conventional data only (Kobayashi et al., 2014). As MERRA-2 data are available after January 1980 and NCEP-CFSR ends

in December 2010, monthly mean data from January 1980–December 2010 are mainly analyzed for the comparison. Data from January 1980–December 2015 are used for MERRA (MERRA ends in February 2016) and data until December 2016 are used for ERA-I, ERA5, JRA-55 and MERRA-2. JRA-55C is available until December 2012.

Monthly mean zonal wind and temperature data analyzed in this study were computed from daily means. Daily mean data are the average of the instantaneous 00, 06, 12 and 18 UTC analyses in ERA-I, JRA-55, JRA-55c and NCEP-CFSR. In

ERA5, the daily means are calculated from instantaneous hourly data, from 00 to 23 UTC. For MERRA and MERRA-2, daily means consisting of instantaneous 3-hourly 'asm' output are used (see more details for 'asm' in





https://gmao.gsfc.nasa.gov/reanalysis/MERRA-2/docs/ANAvsASM.pdf). Consequently, the solar diurnal (24-hour) and semidiurnal (12-hour) tides should be eliminated in monthly mean data in each case, but the effect of higher order tides (e.g., 8-hour, 6-hour) could still be present in the monthly means for those reanalyses with 6 hourly instantaneous data. However, the effects of these tides on the zonal mean should be extremely small, at least at altitudes analyzed in this study (up to 1 hPa

for all reanalyses comparison and 0.1 hPa for the MERRA vs. MERRA-2 comparison).

In global models, sponge layers are commonly set near the upper boundary in order to reduce unrealistic reflection of vertically propagating waves from the model top. The formulation of sponge layers differs among each reanalysis operational model.  Wright et al. (https://jonathonwright.github.io/pdf/S-RIPChapter2E.pdf) summarized the details of sponge layers and the placement of the model top level. The model tops are ~0.266 hPa in NCEP-CFSR, 0.1 hPa in ERA-I,

and JRA-55, and 0.01 hPa in ERA5, MERRA and MERRA-2. Sponge layers in ERA-I and ERA5 are applied above 10 hPa by adding an additional function to the horizontal diffusion terms, whose strengths varies with wavenumber and model level. In addition, ERA-I includes Rayleigh friction but ERA5 does not. JRA55 includes a sponge layer which gradually enhances horizontal diffusion coefficient with height above 100 hPa and Rayleigh friction is also used above 50 hPa. MERRA and MERRA-2 implement sponge layers above ~0.24 hPa by increasing the horizontal divergence damping coefficient. NCEP-

CSFR applies Rayleigh damping above ~2 hPa in addition to employing a height dependent horizontal diffusion coefficient. Indeed, the treatment of sponge layers differs quite considerably among the reanalysis models.

We note here some particular concerns noted earlier about the tropical middle atmosphere representation in the MERRA-2 reanalyses. The dynamical model used in producing the MERRA-2 reanalyses is able to simulate a spontaneous QBO in the tropical stratosphere because it includes strong parameterized momentum fluxes from non-orographic gravity waves (Fig. 3

of Molod et al., 2015). Kawatani et al. (2016) showed that MERRA-2 exhibits spurious semi-annual variations of the 10 hPa zonal wind in the 1980s and in late 1993. The downward propagation of the westerly SAO phase is unrealistically enhanced during these periods, presumably because of overly strong gravity wave-forcing (Fig. 3 of Molod et al., 2015). Coy et al. (2016) also noted that MERRA-2 appears to overemphasize the annual cycle before 1995.

The representation of the QBO and SAO in ERA5 has also been discussed recently in conferences and informal reports

(https://confluence.ecmwf.int/display/CKB/ERA5%3A+The+QBO+and+SAO; Shepherd et al. 2018). ERA5 data above 1 hPa are not currently available publicly, but these recent investigators did have access to ERA5 at higher levels for the limited period 2008-2017. They concluded that the QBO at altitudes from about 50 to 5 hPa and the SAO from 5 to 0.5 hPa in ERA5 are close to those in ERA-I. However, the representation of the SAO above 0.5 hPa is very different between ERA5 and ERA-I. Shepherd et al. (2018) reported that the operational global model used to generate ERA5 simulates

unrealistically strong westerlies (spurious equatorial mesospheric jet) around 0.1 hPa during October and May.



For comparison with the reanalyses we will use the two zonal mean temperature and wind datasets derived by Smith et al. (2017), one for January 2002–December 2016 based on SABER measurements, and one for August 2004–December 2016 based on MLS measurements. Note here that MERRA-2 is the only reanalysis that assimilates temperature data from Aura MLS but only at pressures less than 5 hPa, and none of the reanalyses assimilate SABER data (Fujiwara et al. 2017).

The various reanalysis and satellite datasets are provided on a variety of horizontal grids and vertical pressure level structures. We have interpolated the winds and temperatures from all the datasets onto a common 1.5° longitude-latitude grid on41 standard pressure levels from 1000 to 0.1 hPa. Note that MERRA and MERRA-2 provide data on pressure levels up to 0.1 hPa, while the others provide pressure level data only up to 1 hPa. In addition, however, model level data above 1 hPa are available for the ERA-I, JRA-55 and JRA-55C reanalyses, and we interpolated these data to pressure levels allowing
extension of the data to 0.1 hPa. Only data up to 1 hPa are analyzed here for NCEP-CFSR and ERA5.

**3 Differences of the overall patterns of tropical zonal wind and temperature among reanalyses**

Figure 1 shows the time-height variation of monthly mean zonal mean zonal wind over the equator derived from SABER and MLS observations and in each reanalysis from 2002–2016. All reanalyses clearly capture the basic features of the QBO, including the cycle-to-cycle variation in period and amplitude. The SAO is also represented in all reanalyses, and in each
case the SAO zonal wind is qualitatively similar to that derived from the satellite observations. It is evident, however, that the differences among reanalyses are more pronounced at the SAO region than in the lower and middle stratosphere. The substantially weaker amplitude of the SAO in JRA-55C compared to that of JRA-55 indicates the importance of satellite data for the representation of the SAO. In the rest of this section, results of JRA-55C are omitted in order to compare reanalyses under the same condition (i.e., reanalyses assimilating satellite data).

Figure 2 displays vertical profiles of the climatological annual and zonal mean zonal wind and temperature over the equator. Climatology is calculated from January 1980 to December 2010 for the reanalyses, whereas it is calculated for more recent years for the two satellite datasets after SABER and MLS data were available (from January 2002 and from August 2004, for SABER and MLS, respectively, until December 2016). For temperature, reanalyses agree well with the satellite climatology and differences among reanalyses are relatively small (The exception is JRA-55 above ~0.5 hPa where it is an outlier; the
model used in the JRA-55 assimilation is known to have an artificial sponge layer above 1 hPa). In contrast, the spread among the reanalyses is quite large for the equatorial zonal wind. MERRA-2 shows a westerly bias compared to other reanalyses and observation above 20 hPa. Above 10 hPa, differences among both reanalyses and anomalies from SABER and MLS become significantly larger. Above 0.5 hPa, zonal wind in all reanalyses trends toward zero, even for MERRA and MERRA-2, while satellite observations represent stronger climatological westerlies. The long-term mean of the satellite
zonal winds showing mean easterlies in the middle stratosphere and westerlies in the lower mesosphere is in good overall agreement with that computed from rocketsonde observations at low latitudes (Hitchman and Leovy, 1986).



To quantify the spread among reanalyses, the standard deviations (SD) among the reanalyses are calculated as follows:

$$SD = \sqrt{\sum_{i}^{N} (u_i - [u])^2 / N} \; , \tag{1}$$

where $i$ represents the individual datasets from among the $N$ datasets included. The square brackets [ ] denote the mean over all $N$ reanalyses. The SD is calculated for each month using the monthly mean zonal wind. We calculate the SD among all

six reanalyses from 1980 to 2010, the SD among the five reanalyses (excluding NCEP-CFSR and extended until 2015) as well as the SD between the two MERRA versions (MERRA and MERRA-2) from 1980 to 2015. The SD among both six and five reanalyses is discussed up to 1 hPa whereas the SD between MERRA and MERRA-2 is discussed up to 0.1 hPa (i.e., the maximum altitude of pressure data provided).

Figure 3 shows the time series of monthly mean zonal mean equatorial zonal wind in the satellite derived datasets (black

lines) and in each reanalysis plotted together with the zonally averaged SD among the reanalysis at 1 and 0.1 hPa. Solid lines in Figs. 3c and 3d are the 1-year running mean of the SD. At 1 hPa, all reanalyses (ERA-I, ERA5, JRA55, MERRA, MERRA2 and NCEP-CFSR) represent the SAO with qualitative agreement with the satellite-derived winds. However, there are substantial differences among the reanalyses in the individual monthly values (SD is typically ~3-20 m s$^{-1}$) MERRA-2 sometimes represents significantly stronger westerly extremes than other reanalyses (e.g. during 1980 and 1989). ERA-I and

ERA5 also sometimes have stronger westerly extremes (e.g. in 1991 and 1996). At 0.1 hPa the disagreement among reanalyses (ERA-I, JRA55, MERRA and MERRA-2) is larger (typically ~5-25 m s$^{-1}$). MERRA winds are much larger easterlies than MERRA-2 until ~1998 and these easterly extremes become smaller later on, as in MERRA-2, although the SAO phases between MERRAs are quite different (see also Fig. 9 shown later). At 0.1 hPa the winds derived from both the SABER and MLS satellite observations have a strong disagreement with all the reanalyses which display much weaker

westerlies in the annual, long-term mean

The time-height section of the zonal mean SD of the equatorial zonal wind and temperature among the five reanalyses and then just between MERRA and MERRA-2 is shown in Fig. 4. Both zonal wind and temperature SDs decrease with time, a trend that is particularly clear at levels from 70 hPa to ~3 hPa. As discussed in Kawatani et al. (2016), one possible reason for the reduction of the SD over time is the upgrading of satellite radiance observations. From 1979 to 2006, the TOVS

{TIROS (Television InfRrared Operational Satellite) Operational Vertical Sounder} Stratospheric Sounding Unit (SSU), and Microwave Sounding Unit (MSU), were available. After May 1998, data from the Advanced Microwave Sounding Unit (AMSU) became available. Satellite radiance data will presumably affect the assimilated temperatures in the stratosphere, but will also have considerable influence on the wind (cf. Iida et al., 2014). Near the equator thermal wind shears are particularly sensitive to small errors in observed temperatures. After 2000, the amount of available satellite data increased

greatly (e.g. Kobayashi et al., 2015), and the contributions of satellite radiance data to better representation of the tropical





winds also presumably increased. Kawatani et al. (2016) also showed the evolution of the number of available monthly mean near-equatorial radiosonde observations at 10–70 hPa. The number of available radiosonde observations generally increased with time at all levels, which also likely contributed to the decreasing trend of SD among reanalyses from 70–10 hPa.

Next, we explore spatial variation of SD among reanalyses. Figure 5 displays the latitude-height distributions of the zonal means of the zonal wind and temperature SD among six reanalyses averaged from 1980 to 2010, as well as the SD between MERRA and MERRA-2. In the SD among the six reanalyses (Figs. 5a and 5b), the largest values are on the equator and increase from the upper troposphere to the upper stratosphere so that the largest values occupy a wedge-shaped region shown clearly in the figure. The temperature SD shows an opposite structure, having a minimum SD in the lower latitudes and becoming larger at higher latitudes. Near the equator small differences in temperature (say between two reanalysis datasets) can be expected to result in large differences in the thermal wind shear. The zonal wind and temperature SDs between MERRA and MERRA-2 (Figs. 5c and 5d) show qualitatively similar structures. The local maximums of zonal wind SD around 50°S and 50°N at 0.1-0.3 hPa are due to the different shape of the mesospheric jets between MERRA and MERRA-2 (not shown). This may result from the inclusion of parameterized non-orographic gravity wave drag in MERRA-2. The observational constraints in the lower mesosphere are much weaker and the representation of the zonal winds may depend strongly on the model configuration used in each reanalysis.

Figure 6a shows the horizontal distributions of the zonal wind SD among six reanalyses at 1 hPa. The SD shows a fairly zonally uniform structure. Kawatani et al. (2016) showed the zonal wind SD from 70 to 10 hPa, and noted that the SD becomes more zonally uniform as height increases. The 0.1 hPa SD between the MERRAs also shows a fairly zonally uniform structure (Fig. 6b). Figures 6c and 6d display the zonal wind SD after the zonal mean of the SD is subtracted. This shows that there actually are some systematic zonal variations in the SD. Notably positive SD anomalies are found in the middle Pacific, in the lower to middle stratosphere, where *in situ* observations are few (Fig. 5 of Kawatani et al., 2016). The smallest SD is seen near Singapore (1.4°N, 104°E), where high-quality radiosonde observations up to 10 hPa are consistently available (Naujokat, 1986). It is interesting that the region of reduced SD (negative SD anomaly in Fig. 5c) extends up to ~3 hPa. This suggests that the influence of *in situ* observations near the equator on the reanalyses extends to altitudes considerably above the actual observation heights. At 1 hPa, the SD in the eastern hemisphere is slightly smaller than that in the western hemisphere. Another lower SD over South Africa around 50°W does not extend as high as it does over the Singapore, because the observational density at 10 hPa is significantly lower over South America, and the higher density is limited at 50–70 hPa (Fig. 5 in Kawatani et al., 2016). The zonal wind SD between MERRA and MERRA-2 (Figs. 6b,d) also show a similar structure; however, the reason for the sudden increase of the zonal asymmetry of the SD at ~0.5 hPa is unclear.





## 4 Similarity and differences of the semiannual oscillation among the reanalyses

In this section, we focus on the similarity and differences of the seasonal cycle (dominated of course by the semiannual component) among reanalyses. Figure 7a shows the long term mean annual cycle of zonal mean equatorial zonal wind at 1 hPa as derived from SABER and MLS observations, and in each reanalysis. Some differences are seen between the winds

based on SABER and MLS, e.g. stronger westerly maxima (around April and October) and weaker easterly maxima (around January and July) in the MLS-derived winds compared to the SABER-derived winds (consistent also with the long-term annual mean equatorial zonal wind shown in Fig. 2a). The reanalyses zonal winds shown in Fig. 7a in most months are within ~10 m·s$^{-1}$ of one of the satellite-derived values, with the exception of MERRA-2 which has clear westerly biases in all months compared to satellite-derived winds and actually displays westerlies for the climatological values in July (i.e. during

the second easterly phase of the SAO as shown in other datasets).

Figure 7b shows a measure of the interannual variability of equatorial zonal mean zonal wind in observations and reanalysis. This variability is characterized by the SD for each calendar month. For example, January SD in each reanalysis is calculated from the monthly mean zonal wind data from 1980 to 2010 (31 data points). Note that the analyzed periods for the satellite derived winds are shorter than those of the reanalyses (after January 2002 and August 2004 for MLS and SABER,

respectively, until December 2016); reanalyses generally have a larger SD compared to satellite-derived winds. The SD in MERRA-2 varies significantly among months, being larger during the easterly phase of the SAO than during the westerly phase. The interannual SD of ERA-I, ERA5, JRA-55 and NCEP-CFSR show little variability through the year as do the satellite-derived winds from both SABER and MLS.

Figure 8 shows the time-latitude cross sections of the climatological annual cycle of the zonal mean zonal wind at 1 hPa for

induvial datasets (in the rest of this section, results of JRA-55c are included again to compare it to JRA-55). The patterns are similar between satellite-derived winds based on MLS and SABER as already shown in Smith et al. (2017, their Fig. 5). The easterly extends into the tropics from the summer hemisphere in the solstices; the westerly exists in all latitudes during the equinox season. This characteristic is qualitatively represented by all reanalyses. MERRA-2 features significantly stronger equatorial westerlies around both equinoxes. This can be explained by the fact that the westerly phase of the SAO is believed

to be driven mainly by the atmospheric waves (Dunkerton, 1982), and the dynamical model used in MERRA-2 includes quite strong parameterized momentum fluxes from non-orographic gravity waves. In contrast, JRA-55 and NCEP-CFSR represent weaker westerlies during equinoxes. Comparing JRA-55 to JRA-55C, both easterlies and westerlies are significantly weaker in JRA-55C, once again demonstrating the significant contribution of satellite temperature observations to the representation of a realistically strong SAO.

Figure 9 shows time-height cross sections of the climatological mean annual cycle of the zonal mean zonal wind over the equator in each of the datasets. The satellite-derived winds from SABER and MLS (Figs. 9a,b) agree fairly well, particularly



below about 0.3 hPa and these results agree reasonably well with the time-height section of the SAO determined from rocketsonde observations (Garcia et al., 1997; Smith et al., 2017). The equinoctial maxima of the westerly peak are above 0.1 hPa; solstices while the easterly wind peaks are at ~1 hPa. The satellite derived winds clearly show that the SAO cycle in the first half of the year (boreal winter and spring) is stronger than that in the second half (austral winter and spring), a

feature that has been apparent since the earliest rocketsonde studies of the SAO (Reed, 1966). All of the reanalysis zonal wind datasets differ considerably from the satellite-derived winds and the reanalyses differ significantly among themselves as well. The SAO westerlies in the SABER and MLS derived data penetrate up to ~5–7 hPa in the 1st cycle of the year and ~3 hPa in the 2nd cycle. The extent of the westerlies is slightly higher in ERA-I, ERA5, JRA-55, JRA-55C and NCEP-CFSR. This westerly penetration around 5 hPa is well represented in MERRA, while MERRA-2 represents a more downward

penetration of the westerly. MERRA-2 is the only dataset considered that does not have easterly mean winds at 1 hPa in July and August.

At 0.1 hPa, both the SABER and MLS derived equatorial zonal winds (Fig. 9a,b) are westerly throughout the year, while both MERRA and MERRA-2 show easterly phases, although the timing of easterly appearance is different (December and June for MERRA and April and October for MERRA-2). ERA5 data above 1 hPa are not currently available publicly, but

Shepherd at al. (2018) did have access to ERA5 at higher levels for the period 2008-2017 and they note that ERA5 has westerlies throughout the year in the long term mean wind field at 0.1 hPa averaged over 5°S–5°N, which is more similar to the SABER and MLS derived winds and very different from ERA-I (Fig. 9c).

Figure 10 is the same as Fig. 9, except long-term annual mean climatology (i.e. the mean zonal wind in Fig. 2a) has been removed at each level. In this figure, the SAO easterly and westerly phases are more clearly visible. A stronger SAO easterly

in the 1st cycle compared to the 2nd cycle is seen in both observations and all reanalyses. Differences among reanalyses are still obvious in this figure.

To investigate the SAO in more detail, SAO components were extracted from each of the datasets considered here and then the SAO amplitude was calculated as follows:

$$SAO\,amplitude = \sqrt{2} \times \sqrt{\sum_{i}^{N} u'^{2}_{i} / N} \;, \tag{2}$$

where $u'$ is the monthly mean zonal wind of the filtered SAO component, and the sum is over all the months in the time series for each dataset.

Figure 11 shows the latitude-height cross section of SAO amplitude in zonal wind. The observed SAO amplitude has its maximum off the equator in the Southern Hemisphere, around 7.5° S in the SABER-derived dataset and 12°S in the MLS-derived dataset. These results are in basic agreement with the earlier study of Hopkins (1975) who analyzed the SAO





amplitude determined from roughly 10 years of rocketsonde data at several stations and concluded that the SAO amplitude near the stratopause peaked around 10-15°S. The SAO amplitude from the MLS-derived winds has a more pronounced asymmetry between the northern and southern hemispheres, notably with the region of large amplitude near the stratopause appearing thinner on the northern side of the equator, again reminiscent of the pattern in Hopkins (1975) result. Note again

that the near-equatorial winds in our satellite-derived datasets were estimated by cubic spline interpolation of the balance winds at and poleward of latitude $\pm$ 8° for SABER and $\pm$6° for MLS (Smith et al., 2017) and this must introduce uncertainty in the wind estimates near the equator in each case.

The results for the SAO zonal wind amplitude for each of the reanalyses (Figs. 11c-i) show the overall pattern of a peak near the low latitude stratopause, but the details differ quite substantially among the different datasets. Notably the degree of

10 interhemispheric asymmetry differs; the maximum latitude of the SAO amplitude at 1 hPa is at 3°S in ERA-I, 4.5°S in ERA5, 12°S in JRA-55, 3°S in JRA-55C, 4.5°S in MERRA, 9.5°S in MERRA-2, and 13.5°S in NCEP-CFSR. The peak SAO wind amplitude is larger than in the satellite-derived datasets (Figs. 11a,b) in ERA-I, ERA5, and MERRA, but smaller in NCEP-CFSR. The SAO peak amplitude is much weaker, and occurs at a lower altitude, in the JRA-55C versus the JRA-55 reanalyses. Poleward of 20° in both hemispheres, the SAO amplitude becomes larger with latitude (e.g. from 20° S–30° S

and/or 20° N–30° N, from 0.3 to 1 hPa) in both MERRA and MERRA-2 (Figs. 11g,h), features which find no support in the corresponding result from the satellite-derived datasets (Figs. 11a,b).

Figure 12 shows the same amplitude estimate as in Fig. 11, but for the SAO in temperature. In contrast to the zonal wind, the temperature SAO amplitude is equatorially-centered and appears distributed more symmetrically about the equator. Different latitudinal gradients of the temperature, which are larger in the Southern Hemisphere, result in a southward shift of the

20 maximum zonal wind amplitude in the satellite-derived datasets. The maximum SAO amplitude of temperature over the equator is found at 2–3 hPa, with another peak at 0.1 hPa in both SABER and MLS. Large SAO components are also found at 40° S and 40° N at 0.1 hPa in both the SABER and MLS observations.

The temperature SAO amplitude at 2 hPa over the equator is approximately 4.3 K (SABER), 4.2 K (MLS), 5.6 K (ERA-I), 4.7 K (ERA5), 4.0 K (JRA-55), 2.2 K (JRA-55C), 4.2 K (MERRA), 3.9 K (MERRA-2), and 2.9 K (NCEP-CFSR). ERA-I

and ERA5 overestimated the temperature SAO amplitude compared to the satellite observations, while the amplitude is significantly underestimated in NCEP-CFSR. The much weaker SAO apparent in JRA-55C than JRA-55 (Fig. 11f versus Fig. 11e) again demonstrates the key role assimilated satellite data play in defining the SAO in reanalyses. For the second amplitude maximum over the equator at 0.1 hPa MERRA has a larger peak value and meridionally wider structure compared with the amplitude computed from the satellite datasets, while MERRA-2 significantly underestimates this maximum. The

0.1 hPa equatorial maximum seen in the satellite datasets is not apparent in the JRA-55 reanalyses and is quite weak in the ERA-I reanalyses. These discrepancies may be attributable to the sponge layers with strong numerical dumping of wave





components above 1 hPa for ERA-I and JRA-55. The large SAO amplitudes near 0.1 and 40°S and 40°N have no counterpart in any of the reanalysis datasets except possibly MERRA and MERRA-2 near 40°S.

The amplitudes shown for the reanalysis datasets in Fig. 12 are calculated using all the data from January 1980 to December 2010. Here, it should be noted that MERRA-2 is the only reanalysis that assimilates the MLS temperature above 5 hPa. Thus,

some improvement is expected for MERRA-2 in the SAO temperature field, after the MLS data were assimilated starting in August 2004. Figures 13a and 13b show the same quantity as Fig. 12 but for MERRA-2 in 1980–2003 (i.e., pre-MLS years) and in 2005–2016 (i.e., when the MLS data are included in the assimilation over each entire year). Figure 13c shows vertical profiles of the temperature SAO amplitude over the equator for SABER, MLS, and MERRA-2 in 1980–2003 and in 2005– 2016. It is evident that over ~2-0.5 hPa the MERRA-2 SAO amplitude over the equator become closer to the SABER and

MLS observations after the MLS temperature was assimilated, although the significantly smaller amplitude at 0.1 hPa is not improved. Notably, the MERRA-2 representation of the SAO amplitude near 0.1 hPa and 40°S and 40°N improves in the period with MLS data included in the assimilation.

## 5 Summary and concluding remarks

The systematic observation of the equatorial middle atmosphere began with balloon-borne radiosondes in the 1950s,

rocketsondes and radars in the 1960s and satellite radiance measurements in the late 1970s. These observations turned up interesting and unexpected phenomena, notably the occurrence of well-defined, very low frequency oscillations of the zonal mean circulation at various altitudes: the QBO in the lower and middle stratosphere and the SAO with large amplitude in the stratopause region and then again in the mesopause region. The tropical middle atmosphere has its own unique dynamics and is known to be particularly challenging for realistic simulation by comprehensive general circulation models (e.g., Butchart

et al., 2018). Global reanalyses have proven to be very valuable in efforts to understand the dynamics of the atmosphere and the climate system, but the accurate representation of the atmospheric flow in gridded analyses is particularly challenging for the tropical middle atmosphere. As discussed in the introduction of Kawatani et al. (2016), the initial attempts in the 1980s and 1990s at global analyses based on variational assimilation into dynamical model integrations produced results with very poor representation of the tropical stratosphere. More modern reanalyses datasets have produced much more satisfactory

results, at least in the lower and middle stratosphere where a quite realistic QBO is apparent in the analyzed zonal wind and temperature fields. Kawatani et al. (2016) compared the monthly-mean zonal wind field up to 10 hPa in several reanalysis datasets and found that the disagreement among the reanalyses was largest on the equator and grew with height (at least up to 10 hPa).

In the present paper we assessed the representation of the zonal wind and temperature fields near the equator in the region

10-0.1 hPa where the dominant feature is the near-stratopause SAO. One complication of examining this region is the nearly complete lack of relevant *in situ* measurements to compare with. While the SAO was first discovered in rocketsonde wind



measurements, Baldwin and Gray (2005) found that rocketsonde data at stations within 10 degrees of the equator that were frequent enough to produce reasonable monthly mean zonal winds were available only during 1965-1983. There is very little overlap with the reanalysis datasets we evaluated as our interest is confined to reanalyses after 1979 when satellite radiometer data are included in the assimilations. In our project we compared, where possible, the reanalyses to observed

limb-sounding radiometer data from SABER and MLS, using both the temperature retrievals and the analysis of Smith et al. (2017) who applied a dynamical balance relation to compute zonal winds from the geopotentials derived from the SABER and MLS temperatures.

Our study evaluated six major global atmospheric reanalyses datasets that have been widely applied in studies of the middle atmosphere (ERA-I, ERA5, JRA-55, MERRA, MERRA-2, and NCEP-CFSR). All six reanalyses have a good representation

of the QBO in the equatorial lower and middle stratosphere and each displays a clear SAO centered near the stratopause. However, the differences among reanalyses are much more substantial in the SAO region than in the QBO dominated lower and middle stratosphere. We followed Kawatani et al. (2016) in characterizing the degree of disagreement among the reanalyses by the SD of the monthly-mean zonal wind and temperature. The zonal wind SD has an equatorial maximum that increases with height. Above 10 hPa even the long term annual mean equatorial wind differs substantially among the

reanalyses. Above 10 hPa there are also substantial differences in the equatorial zonal wind fields between each of the reanalyses and the two satellite-derived wind datasets.

In their study of low latitude winds below 10 hPa, Kawatani et al. (2016) found substantial zonal variation in the SD characterizing the disagreement among reanalyses. Notably this was tied to the availability of *in situ* balloon measurements: the smallest SD was around the longitude of Singapore where consistently high-quality near-equatorial radiosonde

observations are available and the largest values were around the eastern and central Pacific where there are virtually no stratospheric *in situ* wind observations close to the equator. Kawatani et al. (2016) found that the zonal contrast is reduced as height increases, but that a significant variation is found still at 10 hPa. In the present study we extended this analysis to the 0.1 hPa level. Interestingly the near-Singapore minimum in SD is evident to at least ~3 hPa (Fig. 6), i.e. considerably higher than the usual ~10 hPa ceiling for weather balloons.

Over the equator we showed that there are overall long term trends for both the zonal wind and temperature SD to become smaller and this is seen clearly from 70 hPa to ~2 hPa (Fig. 4). Since within each reanalysis the dynamical atmospheric model and the analysis procedures are fixed, improvement with time must be related to an increase or improvement in the actual observations available for assimilation. The temperature sounders on NOAA operational satellites were upgraded significantly in May 1998, from the SSU to the Advanced Microwave Sounding Unit (AMSU). Kawatani et al. (2016)

showed an improvement around 1998 of the QBO representation up to 10 hPa and a reduction in the wind and temperature SD. The present paper shows that the long term decreasing trend of the SD is seen at least up to 2 hPa.



The latitude-height cross sections of the SD display opposite structures between zonal wind and temperature (Fig. 5). The zonal wind SD has a prominent equatorial maximum, indicating the particularly challenging nature of the reanalysis problem in the low-latitude stratosphere, where the Coriolis parameter is small and *in situ* observations are sparse. The zonal wind SD in low latitudes becomes larger with height, showing a wedge-shaped structure. In the mid-to-high latitudes, the zonal wind

SD becomes smaller. In contrast, the temperature SD has minimum values over the equator and maximum values in the polar region, which might be related to the observational density (i.e., significantly fewer observations in the Polar Regions).

We also examined the representation of the upper stratospheric winds and temperatures off the equator. All the reanalyses represent features of the time-latitude variation of the zonal mean zonal wind at 1 hPa that agree broadly with the satellite-derived datasets (Fig. 8). Notably westerlies are present at all latitudes during the equinox seasons while easterlies are clear

in the summer hemisphere and extend to the equator around the solstices. In addition, the amplitude of the SAO in wind and temperature as a function of height and latitude was computed from each dataset (Figs. 11 and 12). The zonal wind SAO amplitude in each case shows a maximum near 1 hPa and at low latitudes, but the details vary somewhat from dataset to dataset. All the results show that the peak SAO is displaced off the equator somewhat into the southern hemisphere (in agreement with earlier studies using rocketsonde measurements), but the extent of the displacement and the values of the

peak SAO amplitude vary considerably among the reanalyses.

Four of the reanalyses considered here provide data values to the 0.1 hPa level, while two others (CFSR and ERA5) have provided data only to the 1 hPa level. The reanalysis fields above 1 hPa in the JRA-55 and ERA-I dataset have to be regarded with some caution, however, as it is known that the dynamical models used in these assimilations have sponge layers with artificial strong damping above 1 hPa. In general, we find significant uncertainties in the reanalyses in the 1-0.1

20 hPa layer. One example is apparent in Fig. 11 where the amplitude of the zonal wind SAO at the equator and 0.1 hPa is about 2.5 times larger in MERRA than in the ERA-I reanalyses.

The Japan Meteorological Agency created a complementary dataset, JRA-55C, by repeating the assimilation procedure for JRA-55, but without including any satellite observations. We compared several aspects of the SAO in the upper stratosphere between JRA-55 and JRA-55C. Very large differences were found and indeed the stratopause SAO in JRA-55C is very weak

(Figs. 11f, 12f). We conclude that the JRA-55C reanalyses are not useful representations of the SAO in the tropical upper stratosphere but the comparison with JRA-55 shows rather directly the great importance of satellite radiance observations to define the SAO in this region.

To sum up, we have examined of the SAO in the tropical upper stratosphere as represented in sophisticated global reanalysis datasets. The reanalyses are able to represent the basic features of the stratopause SAO but there are large uncertainties even

for the very basic fields (temperatures and zonal winds) that we examined. The reanalyses in the tropical upper stratosphere must be regarded as much less reliable than those for the region below 10 hPa where there are at least some high quality *in*



*situ* observations of wind and temperature near the equator each day. On the positive side we found that up to at least ~2 hPa the reanalyses agreed better among themselves as the satellite radiance data used in the assimilations improved. Next generation reanalyses are now appearing, such as MERRA and MERRA-2, as well as ERA5 and JRA-3Q (Shinya Kobayashi, personal communication, 2020), which extend up to the mesopause. We expect that a comparison of multiple reanalyses could be extended up to at least 0.01 hPa in the near future.

**Data availability.** Reanalyses and satellite data used in this study can be inquired about by contacting the authors.

**Author contributions.** YK and TH designed the study. YK analyzed the data. YK and KH wrote the manuscript. AS calculated the zonal wind in SABER and MLS. MF investigated details of each reanalysis. All authors reviewed and edited the paper.

**Competing interests.** The authors declare that they have no conflict of interest.

**Special issue statement.** This article is part of the special issue "The SPARC Reanalysis Intercomparison Project (S-RIP).

**Acknowledgement.** We express our gratitude to the scientific guidance and sponsorship of the World Climate Research Programme (WCRP) coordinated in the framework of Stratosphere-troposphere Processes And their Role in Climate (SPARC) and the SPARC Reanalysis Intercomparison Project (S-RIP). We acknowledge the reanalysis centers for providing their support and data products. YK was supported by Japan Society for Promotion of Science (JSPS) KAKENHI Grant Numbers JP15KK0178, JP17K18816 and JP18H01286, and by the Environment Research and Technology Development Fund (2-1904) of the Environmental Restoration and Conservation Agency of Japan. MF was also supported by JSPS KAKENHI JP18H01286. TH and YK were supported by JSPS KAKENHI JP16H04052 and 17H01159. YK and KH were supported by the Japan Agency for Marine-Earth Science and Technology (JAMSTEC) through its sponsorship of research at the International Pacific Research Center. The National Center for Atmospheric Research is a major facility sponsored by the National Science Foundation under Cooperative Agreement No. 1852977. The GFD-DENNOU Library and GrADS were used to draw the figures.

**Financial support**. This research has been supported by Japan Society for Promotion of Science (JSPS) KAKENHI Grant Numbers JP15KK0178, JP17K18816, JP18H01286 and by the Environment Research and Technology Development Fund (2-1904) of the Environmental Restoration and Conservation Agency of Japan.



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





# Figures

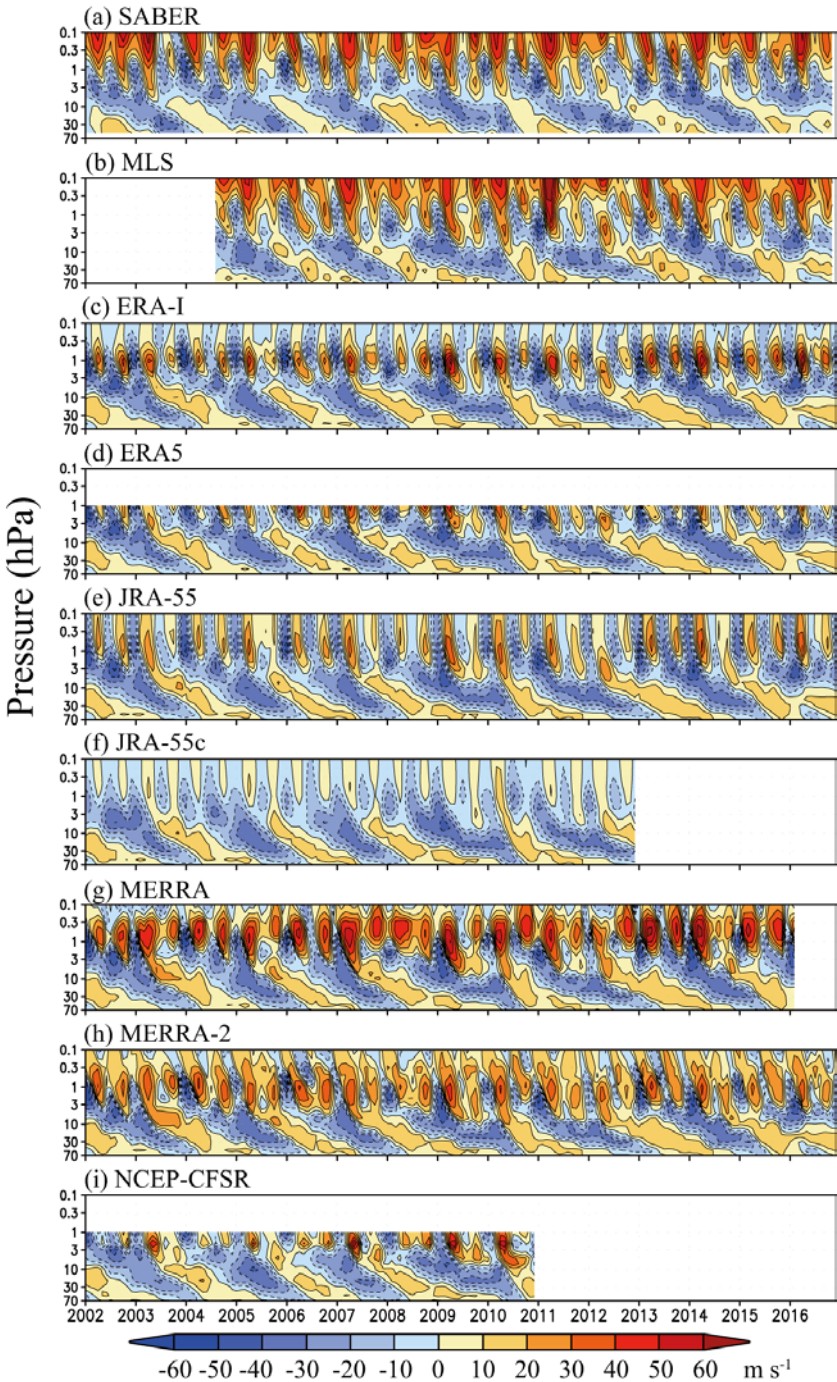

Figure 1: Time-height section of the equatorial zonal mean zonal wind in the (a) SABER, (b) MLS satellite derived datasets as processed by Smith et al. (2017), and (c–i) each reanalysis. The color intervals are 10 m s⁻¹.



Figure 2: Vertical profiles of the climatological annual and zonal mean (a) zonal wind and (b) temperature over the equator for SABER (solid black), MLS (dashed black), ERA-I (blue), ERA5 (light blue), JRA-55 (purple), MERRA (pink), MERRA-2 (red), and NCEP-CFSR (green).

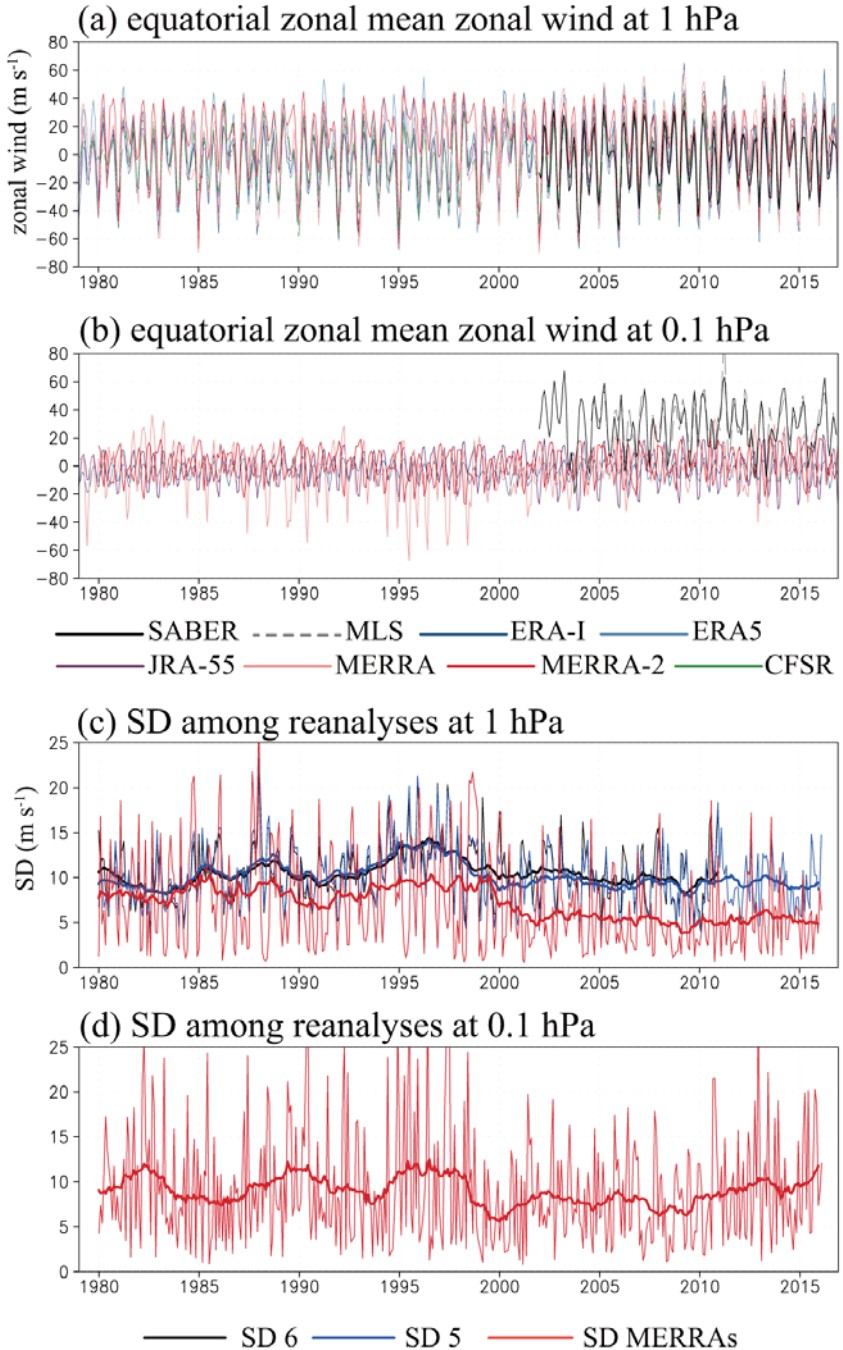

Figure 3: (a,b) Time variations of the equatorial zonal wind at (a) 1 hPa and (b) 0.1 hPa for SABER, MLS, and each reanalysis. (c,d) Zonally averaged equatorial standard deviation among six reanalyses (ERA-I, ERA5, JRA-55, MERRA, MERRA-2, NCEP-CFSR; black), five reanalyses (ERA-I, ERA5, JRA-55, MERRA, MERRA-2; blue), and between
5    MERRA and MERRA-2 (red) at (c) 1 hPa and (d) 0.1 hPa. Thick lines show the 12-month running mean.



Figure 4: Time-height cross section of zonal mean standard deviation of (a, c) zonal wind and (b, d) temperature among five reanalyses (ERA-I, ERA5, JRA-55, MERRA, MERRA-2) and (c, d) between MERRA and MERRA-2 over the equator from 1980–2016. The 12-month running mean is plotted in each case. The color intervals are 0.5, 0.1, 1, 2, 4, 6, 8, 10, and 12 m s[-1] for zonal wind and 0.1, 0.3 0.5, 1, 1.5, 2, 2.5, and 3 K for temperature.





Figure 5: Latitude-height cross-sections of climatological zonal mean standard deviation of (a, c) zonal wind and (b, d) temperature (a, b) among six reanalyses and (c, d) between MERRA and MERRA-2. The color intervals are 0.5, 1, 2, 3, 4, 5, 6, 7, 8 m s$^{-1}$ and K for zonal wind and temperature, respectively. Shading indicates values larger than 0.5 m s$^{-1}$ and 0.5 K.





Figure 6: (Upper) horizontal maps of the zonal wind standard deviation (a) among the six reanalyses at 1 hPa and (b) between MERRA and MERRA-2 at 0.1 hPa; (Lower) longitude-height cross section of anomaly of the zonal wind standard deviation from its zonal mean over the equator (c) among the six reanalyses and (d) between MERRA and MERRA-2. The color intervals are 1 m s⁻¹ for (a, b) and 0.2 m s⁻¹ for (c, d).




Figure 7: (a) Climatological mean annual cycle of zonal mean zonal wind over the equator at 1 hPa for SABER, MLS, and each reanalysis; (b) cycle to cycle variability of equatorial zonal mean zonal wind calculated as the interannual standard deviation in each calendar month.





Figure 8: Time-latitude sections of climatological mean annual cycle of the zonal mean zonal wind for (a) SABER, (b) MLS, and (c–i) each reanalysis at 1 hPa. The contour intervals are 10 m s$^{-1}$.





Figure 9: Time-height sections of climatological mean annual cycle of the zonal mean zonal wind over the equator for (a) SABER, (b) MLS, and (c–i) each reanalysis. The climatology is calculated from January 2002–December 2016 for SABER, from August 2004–December 2016 for MLS, and from January 1980–December 2010 for the reanalyses. The contour intervals are 5 m s$^{-1}$.





Figure 10: Same as Fig. 9 but with annual mean climatological zonal winds (shown in Fig. 2(a)) removed at each level.





Figure 11: Latitude-height cross section of amplitude of the SAO in zonal wind. Color intervals are 2 m s$^{-1}$.





Figure 12: Same as Fig. 11 but for the SAO in temperature. Color intervals are 0.4 K.



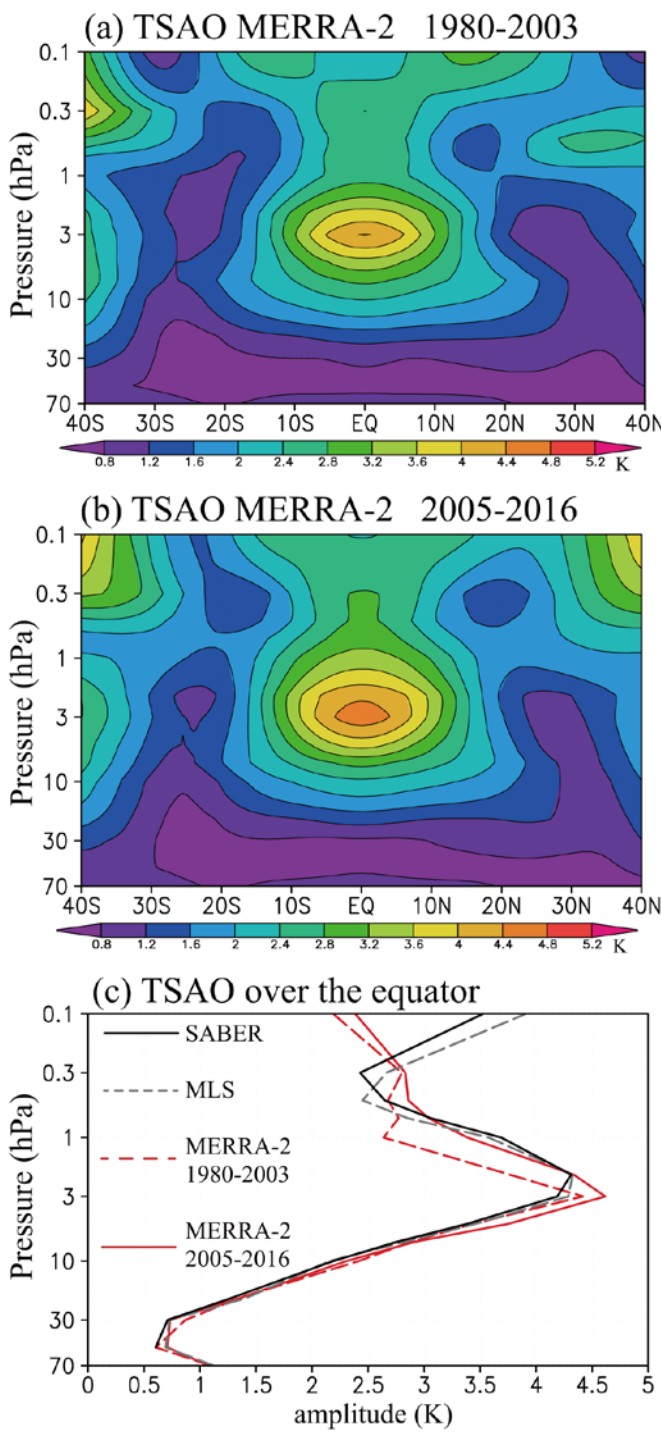

Figure 13: (a, b) Latitude height cross section of SAO temperature amplitude for MERRA from (a) 1980–2003 and (b) 2005–2016; (c) vertical profiles of SAO temperature amplitude over the equator for SABER (solid black), MLS (dashed black), MERRA-2 from 1980–2003 (dashed red), and MERRA-2 from 2005–2016 (solid red).