# Peer review of "Representation of the Equatorial Stratopause Semiannual Oscillation in Global Atmospheric Reanalyses"

_Atmospheric Chemistry and Physics, 2020_

## Referee Comment (RC1) · Gloria Manney (Referee) · 11 Apr 2020

**Review of** "Representation of the Equatorial Stratopause Semiannual Oscillation in Global Atmospheric Reanalyses" by Kawatani et al.

**Recommendation:** Accept with minor revisions

**General Assessment:** This paper does a thorough analysis of how upper stratospheric to lower mesospheric winds and temperatures in the tropics from reanalyses compare with each other and with observational data derived from satellite measurements, focusing on the zonal mean flow and the semi-annual oscillation. Observational data are sparse in this region and our ability to model it is limited, so a paper such as this that quantifies how the reanalyses represent this region and discusses possible reasons for the differences is valuable. The authors do a good job of relating their results to previous literature and our current state of knowledge of this region of the atmosphere. This paper should be suitable for publication in ACP after minor revisions.

**A few small overall comments:**

- a. For many of the "climatological" analyses (that is, those for which averages and standard deviations are calculated over a period of years) there are several different time periods used. Especially, the reanalyses are generally used for 1980--2010, but the SABER and MLS derived fields are used for about 2002--2016 and 2005--2016, respectively. Can you say something about how real atmospheric differences between the two periods may affect the results? Have you looked at reanalysis / satellite comparisons including only the years that are available in both (if that comparison showed significantly different agreement between the reanalyses and the satellite data, it would raise questions about the reasons for disagreements seen in the current comparisons)? (As a minor clarification related to this, since several time periods were used, it would be helpful to list the periods used in each of the figure captions that show climatological fields -- Figures 2, 5--12 -- which is currently done in some, but not all, of these.)
- b. The discussion of the sponge layers in the reanalysis description and ensuing text could be made clearer if you were a little more specific in the former about comparing the vertical regions over which the sponge layers are applied, and differences between the (to use imprecise language) "severity of the damping" at the altitudes you later focus on in the paper. All the information is given, yes, but if you added a sentence or two in the initial discussion of the reanalyses about which ones are more likely to be adversely affected by this at the levels you focus on, it would help the reader follow the thread through the rest of the paper.
- c. I would like to see a little more said about the possible differences related to MERRA-2 assimilation of MLS temperatures at altitudes about the 5hPa pressure level starting in Oct 2005. Figure 13 is useful, as far as it goes, but to what degree (probably a function of latitude given the weak constraint near the equator) would assimilating these temperatures be expected to indirectly (via the underlying model) affect the winds? What about for the other comparisons shown, e.g., the climatological winds and

temperatures? Is agreement between the reanalyses and MERRA-2 significantly improved (or changed in any way) for those fields in the period when MLS data are assimilated?

d. I am a little concerned still about using "balance" winds near the equator, given that I've seen previous results comparing winds calculated thus (using the same balance used by Smith et al, that originally published by Randel, 1987) from analysis/reanalysis GPH with the winds from the analysis/reanalysis system itself that showed very large differences out to 15--20 degrees from the equator at pressures below 50hPa and an expanding latitude region of differences with decreasing pressure, with large disagreement at over 30 degrees from the equator in the upper stratosphere (e.g., see Manney et al, 1996, JGR, 101, 10,311--10334; their Figure 1; recent unpublished results I have using MERRA-2 for this comparison show qualitatively similar results). I realize this is a difficult issue, since we don't have any "truth" field to compare with, but, given that the balance wind calculation done using the GPH from a reanalysis tends to have much lower winds than those from the reanalysis, could you discuss a little more how this may affect your interpretation of reanalysis differences from winds derived from MLS and SABER?

**Specific Comments (in order of appearance in the paper):**

Page 1, Abstract: It would be helpful to say whether there are reanalyses (other than JRA-55C) that do particularly poorly (e.g., CFSR may fit that description for some of the diagnostics?).

Page 3, lines 27--28, could you list Tomikawa et al as "in preparation"?

Page 5, line 26, The ERA5 data on model levels (hence up to altitudes higher than analyzed here) are, and have been for quite some time, publicly available, so this statement is not accurate. However, I know from personal experience over the past year or so that acquiring those fields can be exceptionally painful and time-consuming! I would suggest simply moderating that statement to something like "were not available at the time of this writing" (particularly, leave out "publicly")

Page 5, line 30, Not sure I have access to Shepherd et al (2018) (is there a URL you could give in that citation?), but I'm curious how they know it is spurious (what data do they have to compare to or what physics is it inconsistent with?). Since that reference may not be trivial for every reader to get, could you possibly say a tiny bit more about it?

Page 6, line 25, Related to my overall comment about the sponge layers, don't all or most (I guess MERRA and MERRA-2's start at a bit lower pressure) of the reanalyses have an artificial sponge layer at this pressure? Is JRA-55's more severe?

Page 7, lines 14--15, 18, and Figure 3, It is quite difficult to see these differences in Figure 3, perhaps there is a way to improve this? Might showing differences from a reanalysis mean help?

Page 7, Line 23, This seems to me to be just as clear at lower pressures.

Page 8, lines 26--27, I find this sentence a bit confusing, is "South America" supposed to be "South Africa" or are we talking about two different regions with lower observation density?

Page 8, lines 16--30, Is there a relationship between the asymmetries in SD and asymmetries in the winds and temperatures themselves?

Page 9, line 18, MLS seems to me to show more differences in variability at different times of year, can you comment on this?

Page 9, line 26, could you give a reference for this?

Page 15, lines 2--4, "Next generation" is not appropriate here, especially since one of the reanalyses you list (MERRA) has been superceded (by MERRA-2) and discontinued!

Page 15, line 6, There are DOIs and/or references for the reanalysis datasets themselves, which should be given here. Saying that one can contact the authors for the post-processed data used is appropriate.

**Typos / small corrections:**

Page 1, line 20, add a comma after "Interestingly"

- Page 2, line 2, suggest "The present paper focuses on..."
- Page 2, line 11, add a comma after "waves"
- Page 3, line22, "which" should be "that"
- Page 3, line 21, suggest "in" rather than "by"
- Page 3, line27, "is" should be "are"
- Page 3, line 29 "were" should be "was"
- Page 4, line 2, suggest "At" rather than "In"

Page 4, line 3, "treatments" should be "treatment", "top" should be "tops", and "are" should be "is"

- Page 5, line12, "which" should be "that"
- Page 7, line 13, there is some punctuation (period or colon) missing before "MERRA-2"
- Page 7, line 16, suggest "show much stronger" rather than "are much larger"
- Page 7, Ine 19, add comma after "reanalyses"
- Page 8, line 12, suggest "arise from" instead of "are due to"
- Page 9, line 2, "similarity" should be "similarities"

Page 9, line 22, should be "easterlies extend" and "westerlies exist"

Page 10, line 2, "peak" should be "peaks"

Page 10, line 3, something is wrong here, the word "solstices" seems to be thrown in out of place?

Page 11, line 15, "which" should either be "that" or you could say "for which there is"

Page 10, lines 7--8, this is really unclear because of the use of higher / lower in a way where it is not clear whether it is higher / lower in pressure or altitude. Please re-word this to clarify.

Page 10, line 10, "westerly" should be "westerlies"

Page 12, line5, delete comma after "field"

---

## Short Comment (SC1) · 21 May 2020

Text says : "Consequently, the solar diurnal (24-hour) and semidiurnal (12-hour) tides should be eliminated in monthly mean data in each case, but the effect of higher order tides (e.g., 8-hour, 6-hour) could still be present in the monthly means for those reanalyses with 6 hourly instantaneous data. However, the effects of these tides on the zonal mean should be extremely small, at least at altitudes analyzed in this study (up to 1 hPa 5 for all reanalyses comparison and 0.1 hPa for the MERRA vs. MERRA-2 comparison)."

It is clear that there are two semiannual pulses that operate on the hemispheres

which would lead to a semiannual nodal cycle – see Fig 1 from this presentation from the recent AGU https://presentations.copernicus.org/EGU2020/EGU2020-19821_presentation.pdf

Considering that if this is considered a semi-annual tide, then wouldn't the lunar nodal tide of 13.66 days modulating this tide lead clearly to the 28 month cycle of QBO? See Fig 2. Both the SAO and QBO should show up in global reanalysis
* * *
[Figure]

**Fig. 1.** from https://doi.org/10.5194/egusphere-egu2020-19821

[Figure]

**Fig. 2.** Lunar nodal cycle modulating solar annual cycle

---

## Editor Comment (EC1) · Martin Dameris (Editor) · 29 May 2020

Dear authors,

sorry for the delay!! One referee gave already a full review to your paper in the round of the quick-reviews (technical review); here again are the statements of the reviewer. The referee rated the manuscript in all categories with "excellent". The reviewer ask for minor revisions. Please consider the points in your revised manuscript.

Here is the referees report to your manuscript:

"Representation of the Equatorial Stratopause Semiannual Oscillation in Global Atmo-

spheric Reanalyses" by Yoshio Kawatani, Toshihiko Hirooka, Kevin Hamilton, Anne K. Smith, Masatomo Fujiwara

This paper represents the state of the art in depicting the equatorial stratopause (and mesopause) semiannual oscillation in global reanalyses. Its focus is not on dynamical cause or dynamical diagnosis, but on differences among the reanalyses, in an effort to increase our understanding of "what is the best description of the lower mesosphere and upper stratosphere"? The authors are international experts who have been working intensively on this inter-comparison for several years. The exposition is logical and the technique of comparing standard deviations is helpful for concisely describing differences among data sets. They present helpful detail, both in assessing differences, and in attempting to diagnose the cause of the differences among analyses. I would like to suggest to try adding even more commentary regarding the likely causes of the differences among analyses. In particular, it would be helpful to add a few sentences in the introduction or data and analysis section which describes the altitude range of reliable data for MLS and SABER, and the degree to which they are used in reanalyses, since these are the primary contributors toward improved representation of winds and temperatures in the USLM.

It might be helpful (but certainly not required!) if a summary graph could be included showing an idealized version of the altitude range of reliability for each of the analyses, with specific labelling and footnotes which offer likely reasons for diminished quality. If the authors have access to the following information, it would also be interesting to clarify the degree to which analyses are more strongly regressed toward Singapore radiosondes compared to other tropical stations with comparable accuracy and frequency of launch. I recommend publication with minor revision.

1. l17-18: This basic difference in structure is probably just the latitudinal profile of the Coriolis parameter. It's too bad that there are very few rocket wind profiles to include. That means that differences in low latitudes among analyses may have a lot to do with differences in the manner of treatment of building up heights from temperature

soundings. The fact that SD for temperature is larger in the polar regions may reflect modest differences in sampling of actual large-amplitude Rossby and gravity waves.

2. l19-21: This, and other things that I have read, strongly suggests that there is an embedded preference in algorithms underlying most global analyses which favors Singapore, simply by using a higher weighting factor, compared with other stations. This seems to be a legacy of respect, but should perhaps be relaxed, particularly if there are other reliable radiosonde stations in the tropics (please state them, perhaps near p2. L15-16).

3. l24-27: This is a key theme in the paper. You mention sponge layer differences and the interesting JRA-55 and 55C difference. Do you know of any specific algorithm-based differences in how different satellite data streams are dealt with in different re-analyses?

4. p2, l20-25: Can the influence of HRDI be pointed out in the figures? Are there any lidar temperature profiles that are ever included in global analyses?5. p3, l1-5: When I was doing this for LIMS data, I tried several smoothing techniques for building up Z in the tropics to obtain a good match for zonal winds with rocketsondes. I found that a 1-2-1 smoother in latitude for temperature applied at each level before integrating thick-nesses upward yielded better agreement than smoothing at each level independently. I also tried smoothing more at each level, which degraded the comparison. I also tried smoothing across different ranges of latitude, and decided that within 8s-8n is about right, so as not to include thickness information from the subtropics, spreading inward and upward. I didn't like the results from using a cubic spline, which can yield larger amplitudes at higher altitudes. I only mention this because the growth with altitude of SD among analyses for zonal wind is largest at the equator, making it likely that such simple differences in how smoothing is done may explain quite a bit.

6. p3, l14: Could point out the lack of raobs in the central and eastern pacific.

7. section 2: Several kinds of sources for differences are mentioned, including sponge

layer treatment, and non-orographic gravity waves in MERRA-2. Is it possible to give provide more information about how different satellite data streams are treated in different analyses? On p3, l28 it was suggested that such is not possible, but to whatever extent the authors are aware of helpful information in this regard, please do describe further. Otherwise, it may be time to send in investigative reporters to find out what is in those black boxes, anyway.

8. section 2: If any reanalyses include SABER or MLS data, please describe. If so, can their effect be seen in the figures?

9. p6, description of Figure 1: Are there any other features of interest to point out besides the lack of information above 10 hPa in JRA-55C and increasing disagreement at higher levels?

10. Figure 2: Hitchman and Leovy (1986) summarized what was known from rocket-sondes about the time mean vertical mean profile, which includes time mean westerlies in the lowest stratosphere, easterlies in the middle, and westerlies in the lower mesosphere, as shown here. The differences in time mean profiles in the MS shown here could be more strongly emphasized as a theme (cf. Fig. 3b). The figure 2 caption and figure need to be reconciled (there are two dashed profiles). It is very hard for this reviewer to distinguish the differences among the colors chosen for profiles. Please try to distinguish profiles more clearly, perhaps with dash-dot or thickness variation.

11. p7, discussion of Fig. 4: I don't see a special change in 1999. Please discuss why you include the MERRA panels.

12. p7, l26: "geographical" suggest variation in (x, y) to me. Perhaps "variation in the meridional plane"?

13. p8, l2: The midlatitude maximum in SD for zonal wind may be related to the climatological mean maximum location for the polar night jet (cf. Fig. 6b).

14. Fig. 7: Again, there are two dashed lines but only one is described in the legend.

[Figure]

15. p11, l22: Which other analyses include MLS? Do any include SABER? Please clarify early in the paper.

16. p12, l1: Please weigh in with an editorial decision. My understanding is that an apostrophe takes the place of missing letters, such as in the word "doesn't", or indicates possession, but that plural never has an apostrophe, so it makes reasonable sense to write "1900s" instead of "1900's". Yet "1900's" is quite commonly used.

17. p13, l11-12: This also seems to indicate a very strong regression coefficient for Singapore winds in many algorithms.
* * *

---

## Author Comment (AC1) · 30 May 2020

We Thank the editor's comments. Please find attached our reply.

Please also note the supplement to this comment: https://www.atmos-chem-phys-discuss.net/acp-2020-73/acp-2020-73-AC1-supplement.pdf

---

## Author Comment (AC2) · 19 Jun 2020

We thank you for your comments. Your question about lunar nodal tide and the QBO would be interesting but is not be related with our current study. So we cannot post our comments on this.

---

## Author Comment (AC3) · 21 Jun 2020

**Reply to the queries and comments of Referee 1, Dr. Manney**

We very much appreciate the Reviewer's efforts in considering our manuscript and making suggestions for improvement. In our detailed reply below, we reproduce *the reviewer's comments in blue italics*, while our replies are in a standard font.

**Recommendation:** Accept with minor revisions**

General Assessment: This paper does a thorough analysis of how upper stratospheric to lower mesospheric winds and temperatures in the tropics from reanalyses compare with each other and with observational data derived from satellite measurements, focusing on the zonal mean flow and the semiannual oscillation. Observational data are sparse in this region and our ability to model it is limited, so a paper such as this that quantifies how the reanalyses represent this region and discusses possible reasons for the differences is valuable. The authors do a good job of relating their results to previous literature and our current state of knowledge of this region of the atmosphere. This paper should be suitable for publication in ACP after minor revisions.

We appreciate the Reviewer's positive overall response to our work and for the valuable suggestions provided.

**A few small overall comments:**

a. For many of the climatological analyses (that is, those for which averages and standard deviations are calculated over a period of years) there are several different time periods used. Especially, the reanalyses are generally used for 1980--2010, but the SABER and MLS derived fields are used for about 2002--2016 and 2005--2016, respectively. Can you say something about how real atmospheric differences between the two periods may affect the results? Have you looked at reanalysis / satellite comparisons including only the years that are available in both (if that comparison showed significantly different agreement between the reanalyses and the satellite data, it would raise questions about the reasons for disagreements seen in the current comparisons)? (As a minor clarification related to this, since several time periods were used, it would be helpful to list the periods used in each of the figure captions that show climatological fields -- Figures 2, 5--12 -- which is currently done in some, but not all, of these.)

We have looked at reanalyses and satellites comparisons including only data from 2005 to 2010 (i.e., the period of full overlap between two satellite observations and all six reanalyses). We confirmed that overall characteristics of climatology discussed in this paper are qualitatively similar between 1980–2010 and 2005–2010. In addition, there is no long-term "observation" in 1980-2004 for comparison.

However, we think our main conclusions are not significantly affected by the period considered.

As neither SABER nor MLS temperature were assimilated in reanalysis models (except above 5 hPa in MERRA-2 after 2004), the climatological differences between the reanalyses during 1980–2010 and 2005–2010 should not be influenced by the availability of these satellite data (except for MERRA-2).

Here we show one example; vertical profiles of the climatological annual mean and zonal mean zonal wind and temperature in 2005-2010 are shown in Fig. 2R (the same as Fig. 2 but for different periods). MERRA zonal winds around 0.1–0.5 hPa in 2005–2010 are more westerly than in 1980–2010, because of strong easterlies before ~1998 shown as seen Fig. 3. NCEP-CFSR shows westerly above 3 hPa in 2005–2010, opposite to that in 1980–2010. On the other hand, the results for ERA-I, ERA5, JRA55 and MERRA are similar over the two periods. The tendency of the winds to become very weak above ~0.3 hPa is seen in all these reanalyses during both 1980–2010 and 2005–2010, which is the most obvious differences between reanalyses and satellite observations. We also found that in each reanalysis the zonal mean climatological temperatures during 2005–2010 are nearly identical to those averaged over 1980–2010 in the height region of interest here (pressures less than 70 hPa).

The period when there is overlap between both satellite data sets and all six reanalyses is very short (2005-2010), which is not long enough to discuss differences of "climatology" among reanalyses. Here we prefer to select 1980–2010 for the reanalyses intercomparisons as the main purpose of this study is to investigate similarities/differences among reanalyses, which should contribute to the S-RIP project.

In the revision we added an explanation in Section 2 as follows: "Climatological fields are calculated using data from 2002 to 2016 for SABER and from 2005 to 2016 for MLS (when the MLS data are included in the assimilation over each entire year). Note that different time periods are used among reanalyses (1980-2010), SABER (2002–2016) and MLS (2005–2016) for comparison of climatology. We have confirmed that the overall characteristics of climatology discussed in this paper are not significantly affected by the somewhat different periods analyzed".

Please note we use data from January 2005 to December 2016 for MLS in the revised manuscript (rather than from August 2004) in calculating climatology to use only full calendar years in the analysis.

We list the periods used in each of the figure captions as you suggested, thank you for the suggestion.

Figure 2R: The same as Fig.2 but for averaged from January 2005 to December 2010.

b. The discussion of the sponge layers in the reanalysis description and ensuing text could be made clearer if you were a little more specific in the former about comparing the vertical regions over which the sponge layers are applied, and differences between the (to use imprecise language)"severity of the damping" at the altitudes you later focus on in the paper. All the information is given, yes, but if you added a sentence or two in the initial discussion of the reanalyses about which ones are more likely to be adversely affected by this at the levels you focus on, it would help the reader follow the thread through the rest of the paper.

As we have explained in the paper, the description of the sponge layers is well summarized in Table 2.3 in Wright et al. (2017). https://jonathonwright.github.io/pdf/S-RIPChapter2E.pdf which we referred to in section 2.

MERRA and MERRA-2 implement sponge layers above ~0.24 hPa, which is much higher than other reanalyses. Sponge layers in the NCEP-CFSR are above ~2 hPa and their effects may be smaller in the upper stratosphere than in ERA-I, ERA5 (sponge layers above 10 hPa) and JRA-55 (above 50 hPa). However, the methods of imposing the sponge layers differ among reanalyses (Wright et al. 2017), and how the coefficient of horizontal diffusion and/or Rayleigh friction depends on altitude is also

different. So, it would be very difficult to conclude how much the sponge layers affect each reanalysis data set. As ERA-I, ERA5 and JRA-55 provide pressure level data up to 1 hPa, we assume that their sponge layers were designed not to have a strong effect on the representation of the large scale flow up to 1 hPa.

Following your suggestion, we have added text in section 2 as follows:

"It is difficult to quantitatively estimate how different sponge layers affect representations of the circulation among the different reanalyses. Six reanalyses provide pressure level data up to at least 1 hPa, and so we assume that their sponge layers were designed not to strongly affect the representation of the large-scale features of the circulation up to 1 hPa.".

c. I would like to see a little more said about the possible differences related to MERRA-2 assimilation of MLS temperatures at altitudes about the 5hPa pressure level starting in Oct 2005. Figure 13 is useful, as far as it goes, but to what degree (probably a function of latitude given the weak constraint near the equator) would assimilating these temperatures be expected to indirectly (via the underlying model) affect the winds? What about for the other comparisons shown, e.g., the climatological winds and temperatures? Is agreement between the reanalyses and MERRA-2 significantly improved (or changed in any way) for those fields in the period when MLS data are assimilated?

Figure 1 of Long et al. (2017: https://www.atmos-chem-phys.net/17/14593/2017/acp-17-14593-2017.pdf) shows a rather striking jump in MERRA-2 is seen in global mean temperature anomalies when MERRA-2 started to assimilate MLS temperature at pressure less than 5 hPa.

For zonal wind, as we show in Fig. 2R, vertical profiles of equatorial zonal mean zonal wind and temperature in 2005–2010 in MERRA-2 (red line) are similar to those in 1980-2010 (Fig. 2 in the manuscript), which indicates assimilating MLS temperatures above 5 hPa does not significantly affect the climatological zonal wind.

Figure A (a,c,e) is the same as Fig. 13 but including zonal wind SAO in (b,d,f), in addition to latitudinal variations of temperature SAO at 3 hPa (Fig. Ag) and zonal wind SAO at 1 hPa (Fig. Ah), the maximum altitude of the SAO component. As the MLS zonal winds are estimated by the balanced wind relation that contains some uncertainty, we just prefer to compare temperature fields before/after 2005. For latitudinal variations, it is hard to say if slight differences of meridional structures between 1980-2003 and 2005-2016 are due to assimilated satellite data or due to other aspects such as different representation of the climatology in MERRA-2.

Our calculations do not directly address the issue of how the degree of agreement of MERRA-2 with the other reanalyses may change after 2004. However, at least we can say in the last sentence of section 4 that over ~2-0.5 hPa the MERRA-2 SAO amplitude over the equator became closer to the SABER and MLS observations after the MLS temperature was assimilated, although the significantly smaller SAO amplitude at 0.1 hPa is not improved